# Topical miRNA Delivery via Elastic Liposomal Formulation: A Promising Genetic Therapy for Cutaneous Lupus Erythematosus (CLE)

**DOI:** 10.3390/ijms26062641

**Published:** 2025-03-14

**Authors:** Blanca Joseph-Mullol, Maria Royo, Veronique Preat, Teresa Moliné, Berta Ferrer, Gloria Aparicio, Josefina Cortés-Hernández, Cristina Solé

**Affiliations:** 1Rheumatology Research Group, Lupus Unit, Hospital Universitari Vall d’Hebron, Institut de Recerca (VHIR), Universitat Autònoma de Barcelona, 08035 Barcelona, Spain; blanca.joseph@vhir.org (B.J.-M.); maria.royo@vhir.org (M.R.); 2Louvain Drug Research Institute—Advanced Drug Delivery and Biomaterial, Universite Catholique de Louvain, 1200 Brussels, Belgium; veronique.preat@uclouvain.be; 3Department of Pathology, Hospital Universitari Vall d’Hebron, Institut de Recerca (VHIR), Universitat Autònoma de Barcelona, 08035 Barcelona, Spain; teresa.moline@vhir.org (T.M.); bferrer@vhebron.net (B.F.); 4Department of Dermatology, Hospital Universitari Vall d’Hebron, Institut de Recerca (VHIR), Universitat Autònoma de Barcelona, 08035 Barcelona, Spain; mariagloria.aparicio@vallhebron.cat

**Keywords:** lupus cutaneous, lipid-based nanoparticles, miRNA, topical drug delivery, gene therapy

## Abstract

Cutaneous lupus erythematosus (CLE) is a chronic autoimmune skin disorder with limited therapeutic options, particularly for refractory discoid lupus (DLE), which often results in scarring and atrophy. Recent studies have identified miR-31, miR-485-3p, and miR-885-5p as key regulators of inflammation, apoptosis, and fibrosis in CLE skin lesions. This research investigates a novel topical miRNA therapy using DDC642 elastic liposomes to target these pathways in CLE. DDC642 liposomes were complexed with miRNAs (anti-miR-31, anti-miR-485-3p, pre-miR-885-5p) and characterized through dynamic light scattering and Cryo-TEM. Cytotoxicity, cellular penetration, and therapeutic efficacy were evaluated in primary keratinocytes, PBMCs, and immune 3D-skin organoids. miRNA lipoplexes were successfully synthesized with optimized particle size, surface charge, and encapsulation efficiency. These lipoplexes exhibited effective cellular penetration and low cytotoxicity. Anti-miR-31 lipoplexes reduced miR-31 and NF-κB levels while increasing *STK40* and *PPP6C* expression. Pre-miR-885-5p lipoplexes elevated miR-885-5p levels and downregulated *PSMB5* and NF-κB in keratinocytes. While anti-miR-485-3p lipoplexes reduced T-cell activation markers. Anti-miR-31 and pre-miR-885-5p lipoplexes successfully modulated inflammatory pathways in 3D-skin CLE models. miRNA lipoplexes represent promising candidates for pioneering topical genetic therapies for CLE. Further studies, including animal models, are necessary to validate and optimize these findings.

## 1. Introduction

Cutaneous lupus erythematosus (CLE) is a chronic autoimmune skin disorder with an estimated prevalence of 73 per 10,000, reflecting a recent increase in cases [1]. CLE often manifests as the dermatological component of systemic lupus erythematosus (SLE), occurring in approximately 80% of SLE patients [2]. Its clinical presentation is diverse and typically characterized by a chronic and recurrent course. Among CLE-specific subtypes, discoid lupus erythematosus (DLE) and subacute cutaneous lupus erythematosus (SCLE) are the most common [3]. This condition has a substantial societal impact, ranking as the third-leading cause of disability among dermatological diseases. Notably, around 45% of CLE patients experience significant aesthetic concerns that profoundly affect their quality of life, both professionally and socially [4,5].

The first-line treatment for CLE involves topical and/or systemic corticosteroids, often combined with antimalarial agents and stringent sun protection measures [6]. Second- and third-line systemic therapies include methotrexate, retinoids, dapsone, and mycophenolate mofetil or mycophenolic acid [6]. However, more than 30% of DLE cases remain refractory to these treatments, resulting in disfiguring scars, atrophy, and depigmentation [7]. For such refractory cases, alternative therapeutic approaches are urgently needed.

MicroRNAs (miRNAs) play a pivotal role in the immune pathogenesis of CLE lesions. Notably, miR-31 and miR-485-3p have been identified as DLE-specific miRNAs that promote inflammation, keratinocyte apoptosis, and skin fibrosis [8]. Additionally, miR-885-5p, recently recognized as a keratinocyte-specific miRNA common across CLE subtypes, has been implicated in keratinocyte proliferation, leukocyte recruitment, and the production of inflammatory mediators [9]. miRNA-based therapies, including miRNA mimics and anti-miRNAs, have emerged as promising genetic technologies [10]. Several miRNA-based therapeutics have advanced to phase II clinical trials for other conditions, such as TargomiR (a miR-16 mimic) for mesothelioma [11], Cobomarsen (anti-miR-155) for T-cell leukemia/lymphoma [12], and Miravirsen (anti-miR-122) for hepatitis C infection [13].

For skin-related therapies, topical drug delivery offers the advantage of localized treatment with minimal systemic absorption. However, the stratum corneum (SC), the outermost layer of the skin, presents a significant barrier to drug penetration [14]. Recently, elastic lipid-based vesicles have garnered attention for their potential to enhance skin drug delivery [15,16]. These vesicles, similar to conventional liposomes but engineered with enhanced flexibility and elasticity in their lipid bilayer structure, can penetrate the SC’s lipid lamellar regions through hydration or osmotic forces [17]. They have been investigated as carriers for small molecules, peptides, proteins, and vaccines in both in vitro and in vivo studies [18]. A specific elastic liposome, DDC642, has demonstrated efficacy in delivering miRNA molecules, achieving significant topical penetration into the epidermis [19,20]. This technology has been successfully applied in a psoriasis tissue model, providing proof of concept for its potential as a genetic therapeutic approach [21].

This proposal aims to design a novel topical miRNA therapy using elastic liposomes as a genetic therapeutic strategy for CLE skin lesions. The study will evaluate the effectiveness of the DDC642 liposomal carrier in delivering miRNA mimics or inhibitors to target miR-31/miR-485-3p or to upregulate miR-885-5p in cultured primary skin cells. Additionally, the topical delivery system will be tested using immune-competent 3D skin organoid CLE models to evaluate its efficiency in skin penetration and its therapeutic effects on miRNA activity.

## 2. Results

### 2.1. Characterization of DDC642 Liposomes and Their Complexes

The DDC642 liposome and their miRNA complexes (lipoplexes) were characterized using dynamic light scattering (DLS) and Cryo-TEM analysis. The particle size of the DDC642 liposomes ranged from 88 to 107 nm, with a zeta potential between 37.5 and 59.5 mV, consistent with values reported in the literature [19]. Cryo-TEM images revealed the typical round-shaped unilamellar structures of the liposomes, along with bulges and non-circular deformations indicative of their elastic properties (Figure 1A).

miRNA liposome complexes were prepared at various ratios (20:1, 16:1, 10:1, 5:1, and 1:1) using either anti-miR-31/485-3p or pre-miR-885-5p. Following complexation, the particle size significantly increased across all formulations, though remaining below 200 nm (*p* < 0.001, Figure 1B), with a narrow size distribution (polydispersity index < 0.25). A decrease in surface charge was observed, as expected from miRNA adherence to the DDC642 liposomes. However, this decrease was significant only for lipoplexes at a 20:1 and 16:1 ratios for anti-miR-31 (*p* = 0.037 and 0.047, respectively), anti-miR-485-3p (*p* = 0.020 and 0.014, respectively) or pre-miR-885-5p (*p* = 0.002 and 0.010, respectively, Figure 1C). This limited decrease may be attributed to either low encapsulation efficiency or the formation of multilamellar structures, where anti-miRNA molecules are packed between several vesicle layers. To assess stability under physiological conditions, miRNA complexes at ratio 10:1 and 16:1 were incubated at 37 °C in PBS. Their particle size remained stable for up to 52 h, after which it increased significantly to over 250 nm. When stability was evaluated in the presence of 10% FBS, no significant shift in the zeta potential was observed, and it remained stable between +33.5 mV and +48.5 mV.

Encapsulation efficiency (ee%) was evaluated using fluorescence analysis and was found to be approximately 95% across all formulations, except for the 1:1 ratio, which showed significantly lower efficiency at 45–52% (Figure 1D). Cryo-TEM analysis revealed that the presence of anti-miR-31 or anti-miR-485-3p at a 5:1 ratio induced liposome aggregation, forming cluster-like structures (Figure 1E). At 10:1 and 16:1 ratios, mixed lipid bilayers with miRNA superficially compacted were observed, with reduced aggregation between bilayers (Figure 1E).

### 2.2. Cytotoxic Effect and Cell Penetration

The cytotoxicity of DDC642 liposomes and miRNA complexes at ratio 10:1, 16:1, and 20:1 were assessed in the specific cellular targets of skin, human adult primary epidermal keratinocytes (HEKas) and peripheral blood mononuclear cells (PBMCs), isolated from patients with cutaneous lupus erythematosus (CLE) (*n* = 5, Appendix A). A dose-dependent decrease in cell viability was observed with increasing liposomal concentrations in HeKa cells and PBMCs (Appendix A). At doses of 5 µg/mL and 10 µg/mL of miRNA complexes, cytotoxicity in HEKa cells was significantly lower compared to concentration of 15, 20, or 40 µg/mL (*p* < 0.001, Figure 2A). Similarly, in PBMCs, cell viability was maintained at 5 µg/mL and 10 µg/mL doses (Figure 2B). Not significantly, differences were observed between the several ratio of miRNA complex (Figure 2A,B).

To examine the interaction of DiO-labeled liposomes with HEKa cells and PBMCs, they were applied at concentrations of 5 µg/mL and 10 µg/mL and confocal microscopy images were acquired every 15 min over a 180-min period. In HEKa cells, DiO fluorescence was observed in a concentration-dependent manner: 90% of cells exhibited fluorescence within 30 min at 10 µg/mL, whereas the same level required 120 min at 5 µg/mL (Figure 3A). In contrast, PBMCs exhibited a more rapid increase in fluorescence, with 90% of cells showing DiO positivity at 15 min for 10 µg/mL and 30 min for 5 µg/mL (Figure 3B). Phase contrast images were included in Figure 3A to provide additional context regarding the positioning of fluorescence relative to the cells.

While fluorescence was detected in cells, distinguishing between surface-associated fluorescence and true intracellular localization remains challenging. The majority of the fluorescence signal appears in the focal plane above the transmitted light images, suggesting potential surface binding rather than full penetration. To further assess localization, orthogonal views were analyzed, suggesting a possible fluorescence within the cells (Figure 3C). However, without additional surface markers, it is not possible to conclusively determine whether the liposomes have been internalized. Nonetheless, we observed a significantly faster interaction between liposomes and PBMCs compared to HEKa cells (*p* < 0.0001, Figure 3C).

### 2.3. Transfection Efficiency and Modulation of miRNA-Targeted Pathway Regulation

The ability of lipoplexes to modulate miRNA expression in vitro was examined using anti-miR-31, anti-miR-485-3p, and pre-miR-885-5p, targeting specific cellular pathways regulated by microRNAs. We assessed the effects of anti-miR-31 and pre-miR-885-5p in HEKa cells, while anti-miR-485-3p was evaluated in PBMCs isolated from CLE patients (Appendix A). The effects of these treatments were compared to a lipoplex containing a scrambled miRNA control.

HEKa cells were stimulated with UVB to induce overexpression of miR-31 or downregulation of miR-885-5p, as previously described [8,9]. After 6 h, anti-miR-31 lipoplexes were added at a concentration of 10 µg/mL with 20:1, 16:1 and 10:1 ratio. A significant reduction in miR-31 expression was observed after 24 h at 20:1, 16:1, and 10:1 (fold decrease of 4.15, 5.9, and 6.6, respectively, Figure 4A). To achieve a significant increase in miR-885-5p expression in HEKa cells, pre-miR-885-5p lipoplexes a ratio of 10:1 was needed, resulting in a fold increase of 1.99 (Figure 4B).

*STK40* and *PPP6c*, previously identified as targets of miR-31 in keratinocytes and implicated in CLE pathogenesis [8], were evaluated in UVB-stimulated HEKa cells following 24 h of anti-miR-31 lipoplex treatment. Both genes were significantly overexpressed at a 16:1 ratio (fold increase of 2.19 and 5.16, respectively) and a 10:1 ratio (fold increase of 2.59 and 7.87, respectively; Figure 4C). Similarly, *PSMB5* and *TRAF1*, known targets of miR-885-5p in HEKa cells, exhibited significant downregulation after pre-miR-885-5p lipoplex treatment, but only at a 10:1 ratio (fold decrease of 5.84 and 6.02, respectively; Figure 4D). *STK40* and *PPP6c* were also analyzed after pre-miR-885-5p lipoplex treatment, and *PSMB5* and *TRAF1* after anti-miR-31 lipoplex treatment, but no significant differences were found (Appendix A).

The NF-κB pathway, which is inflammation-activated in CLE and modulated by both miR-31 and miR-885-5p, demonstrated a significant reduction in expression after treatment with anti-miR-31 and pre-miR-885-5p lipoplexes at a 10:1 ratio (fold decrease of 5.81 and 1.99, respectively; Figure 4E). Protein levels of phosphorylated NF-κB were also significantly reduced using this ratio (fold decrease of 16.2 and 3.6, respectively; Figure 4F,G). No significant differences in NF-κB expression were observed at a 16:1 ratio (Appendix A).

Anti-miR-485-3p lipoplexes induced a significant reduction in miR-485-3p expression in PMA/ionomycin stimulated PBMCs at a concentration of 10 µg/mL, specifically at 16:1 and 10:1 ratios (fold decrease of 2.22 and 4.08, respectively; Figure 5A). No significant changes in miRNA target expression were observed at a 20:1 ratio (Figure 5A).

MiR-485-3p plays a regulatory role in T-cell activation, a key process in CLE pathogenesis. Following PMA/ionomycin stimulation for 6 h, an increase in CD69 expression, a marker of T-cell activation, was observed in PBMCs isolated from CLE patients. However, treatment with anti-miR-485-3p lipoplexes for 4 h significantly reduced the percentage of CD69^+^CD3^+^ cells only a 10:1 ratio (reduction of 43% CD3^+^CD69^+^, Figure 5B). Furthermore, expression levels of T-cell activation-related genes—*PIK3*, *PRKCD*, *ICOS* and *ZAP70*—were also significantly downregulated in CD3^+^ T cells treated with anti-miR-485-3p lipoplexes (fold decrease of 3.31, 3.30, 9.80, and 4.93, respectively, Figure 5C). Under non-stimulated conditions, no differences were observed between groups (Appendix A).

### 2.4. Lipoplex Penetration Capacity and Therapeutic Efficacy in Immune 3D-Skin Organoids

The penetration capacity of the lipoplexes was evaluated using a T-skin reconstructed human full-thickness skin model derived from EpiSkin. This model consists of a well-differentiated epidermis, formed from normal human keratinocytes, and a dermal equivalent containing human fibroblasts. To generate an immune 3D-skin organoid, the skin organoid was co-cultured with PBMCs isolated from CLE patients using an insert system (Appendix A, Figure 6A). DiO-labeled lipoplexes were incubated with the immune 3D-skin organoid for 4 h, followed by cryopreservation for immunofluorescence analysis. The lipoplexes penetrated the inner layers of the epidermis and localized within keratinocytes but did not infiltrate the dermis or immune cells (Figure 6B).

To assess the therapeutic efficacy of anti-miR-31 and pre-miR-885-5p lipoplexes in the immune 3D-skin organoid, the model was stimulated with UVB radiation for 6 h prior to treatment with the lipoplexes. Gene expression and immunofluorescence analyses, performed 24 h after treatment, demonstrated significant therapeutic effects. Treatment with anti-miR-31 lipoplexes reduced miR-31 expression to 66% of control levels in the immune 3D-skin organoid (Figure 6C). This was accompanied by the upregulation of *STK40* and *PPP6C* (fold increases of 2.20 and 5.87, respectively, Figure 6C) and a significant downregulation of NF-κB expression (fold decrease of 3.33, Figure 6C). In experiments without UVB radiation, no significant differences were observed between control lipoplexes and anti-miR-31 or pre-miR-885-5p lipoplexes, highlighting the requirement of UVB stimulation to observe therapeutic effects.

Similarly, treatment with pre-miR-885-5p lipoplexes resulted in the downregulation of *PSMB5* and *NFKB1* (fold decreases of 2.56 and 2.50, respectively, Figure 6D) and an upregulation of miR-885-5p expression (fold increase of 1.54, Figure 6D). Immunofluorescence analysis revealed decreased NF-κB levels in keratinocytes localized within the inner layers of the epidermis following both lipoplex treatments (Figure 6E,F).

In contrast, treatment with anti-miR-485-3p lipoplexes did not result in significant changes in miRNA or gene expression within the immune 3D-skin organoid. This lack of efficacy may be due to limited penetration into the dermis and inadequate targeting of immune cells (Appendix A).

## 3. Discussion

Cutaneous lupus erythematosus (CLE) is a chronic autoimmune inflammatory disorder characterized by dysregulated microRNAs (miRNAs), which play a pivotal role in disease pathogenesis by regulating key biological pathways in keratinocytes and immune cells. miRNA-based therapies, which can target multiple genes within a single pathway, offer significant potential for addressing inflammatory skin conditions. In this study, we investigated three potential topical therapeutic approaches aimed at modulating miRNA expression for CLE treatment. Through in vitro experiments, we evaluated their cytotoxicity, cellular penetration, and ability to modulate target gene expression. Using an immune 3D-skin model, we demonstrated that elastic liposomes are effective nanocarriers for topical miRNA delivery into the epidermis, providing a promising strategy to mitigate CLE-associated skin inflammation.

Topical miRNA delivery offers a compelling approach by directly targeting regulatory pathways within affected skin tissues [22]. Elastic liposomes, with their unilamellar structure and elastic properties, improve miRNA stability, enhance cellular uptake, and enable deep tissue penetration, key features for targeted therapies in skin disorders [23]. The DDC642 elastic liposome, recently shown to effectively penetrate psoriasis skin lesions [21], was synthesized and characterized in this study. Dynamic light scattering (DLS) and Cryo-TEM analysis revealed that DDC642 liposomes had a size range of 88–107 nm and a zeta potential of 37.5–59.5 mV, consistent with similar lipid-based systems [24]. Cryo-TEM confirmed their unilamellar structure, suitable for stable cellular uptake [25]. Upon miRNA complexation, the liposomes maintained a size distribution below 200 nm, indicating successful miRNA encapsulation without compromising structural integrity. The observed reduction in zeta potential with increasing miRNA-to-lipid ratios (20:1 and 16:1) aligns with previously reported miRNA-lipid complex formation, where surface charge decreases due to miRNA binding [26]. However, limited zeta potential changes at lower ratios (5:1 and 1:1) suggest suboptimal encapsulation efficiency, potentially due to multilamellar vesicle formation or incomplete loading [27]. Encapsulation efficiency was high (~95%) at most ratios, except for the 1:1 ratio, where it dropped significantly to 45–52%, highlighting the importance of liposome concentration [28].

Cytotoxicity assays conducted on human adult primary epidermal keratinocytes (HEKas) and peripheral blood mononuclear cells (PBMCs) indicated minimal toxicity at clinically relevant doses (5 and 10 µg/mL), confirming the biocompatibility of the liposomal system [29]. Differential cellular interaction profiles highlighted target-specific miRNA delivery, with slower interaction in HEKa cells compared to PBMCs, potentially due to differences in membrane composition and uptake mechanisms [30].

Recent studies have emphasized the role of miRNAs in CLE pathogenesis, as they regulate post-transcriptional expression of genes involved in inflammatory and immune pathways [31,32]. miR-31 and miR-885-5p in CLE keratinocytes modulate NF-κB pathway activation [8], while miR-485-3p influences T cell activation [9]. Our results showed that anti-miR-31 lipoplexes effectively inhibited miR-31 expression, achieving 16:1 and 10:1 ratios (liposome:miRNA). Anti-miR-485-3p lipoplexes similarly inhibited miR-485-3p at these ratios. However, pre-miR-885-5p lipoplexes required a higher 10:1 ratio to overexpress miR-885-5p in keratinocytes, consistent with Desmet et al. [20], who observed a similar requirement of an 8:1 ratio. These findings suggest that miRNA inhibition is generally more efficient than overexpression.

In CLE, miR-31 overexpression elevates NF-κB activity by suppressing STK40 and PPP6C, negative regulators of this pathway [33,34]. Our anti-miR-31 lipoplexes not only inhibited miR-31 expression but also upregulated *STK40* and *PPP6C*, reducing phospho-NF-κB protein levels. Similarly, pre-miR-885-5p lipoplexes at a 10:1 ratio inhibited target genes *PSMB5* and *TRAF1* [9], as well as NF-κB protein. In PBMCs, anti-miR-485-3p lipoplexes modulated T cell activation but required a high 10:1 ratio. This underscores the complexity of T cell activation, which involves extracellular stimulatory signals mediated by T cell receptor (TCR) complexes [35]. The signaling pathways involved in TCR activation are multifaceted, with miR-485-3p playing only a partial role [36].

Representative 3D skin models have been developed to study pathological and physiological skin conditions [37]. These models are designed to mimic the native properties of the skin and facilitate direct studies on drug development therapies [38]. In this context, we have developed an immune 3D skin organoid model using PBMCs isolated from CLE patients, building on a psoriatic skin model [39]. This model offers a physiologically relevant system that replicates CLE pathology and enables the precise evaluation of miRNA-based therapies. Anti-miR-31 and pre-miR-885-5p lipoplexes significantly reduced their respective miRNA targets, confirming the efficacy of liposomal miRNA delivery in modulating key CLE pathways. Downregulation of phospho-NF-κB expression further validated the anti-inflammatory effects of these lipoplexes, consistent with the established role of NF-κB in CLE pathogenesis [8,9,40]. However, the limited efficacy of anti-miR-485-3p lipoplexes in reducing T-cell activation markers highlights the need for improved dermal penetration and immune cell targeting. This could potentially be achieved using alternative nanodelivery systems, such as niosomal gels, which enhance dermal delivery [41].

This study highlights the potential of elastic liposomal formulations as a highly effective delivery platform for miRNA-based therapeutics in CLE, achieving significant gene regulation and anti-inflammatory effects. Promising outcomes were particularly evident in keratinocyte-specific modulation using immune 3D skin models. However, further optimization of liposomal delivery systems is essential to enhance dermal penetration and immune cell targeting, particularly for miRNA therapies such as anti-miR-485-3p. Future research should focus on evaluating anti-miR-31 and pre-miR-885-5p lipoplexes as pioneering genetic therapy approaches to mitigate CLE-associated skin inflammation in lupus animal models, aiming to translate these findings into viable clinical treatments.

## 4. Materials and Methods

### 4.1. Materials and Reagents

DOTAP (2,3-dioleoyloxy-propyl-trimethylammonium chloride), DOPE (1,2-dioleoyl-sn-glycero-3-phosphoethanolamine), and cholesterol (Chol) were purchased from Sigma-Aldrich (San Luis, MO, USA). Pre-miR-885-5p (AM17100), pre-miR miRNA negative control #2 (AM17111), anti-miR-31-5p (AM11465), anti-miR-485-3p (AM10799), and anti-miR miRNA inhibitor negative control #1 (AM17010) were obtained from Thermo Fisher Scientific (Waltham, MA, USA). HEPES buffer (1 M, HB) was purchased from Sigma-Aldrich (Bornem, Belgium) and diluted to a 20 mM solution (pH 7.4).

### 4.2. Particle Preparation and Complex Formation

DDC642 liposome and miRNA complexes (lipoplexes) were prepared using the solvent evaporation method described by Desmet et al. [20]. DOTAP, DOPE, and Chol were dissolved in chloroform at a concentration of 10 mg/mL in a ratio of 6:4:2. The components were mixed and the solvent was removed via rotary vacuum evaporation above the lipid transition temperature. The resulting lipid film was hydrated in HB containing 30% ethanol (EtOH). After overnight incubation, vesicles were kept in an ice-cold bath during sonication (10 min, 10-s on/off intervals, 40% amplitude, 750 W). The vesicles were then filtered through a 100 nm polycarbonate membrane filter (Whatman, Brentford, UK). Corresponding lipoplexes were prepared by diluting the pre- or anti-miRNA molecules with HB to a final volume of 100 μL. Liposomes were added under vortex mixing to achieve liposome:miRNA ratios of 20:1, 16:1, 10:1, 5:1, or 1:1 (*w:w*). To characterize the different conjugates, samples were analyzed using Cryo-TEM, dynamic light scattering (DLS), and zeta potential measurements. DLS and zeta potential measurements were performed using a Zetasizer Nano series instrument (Malvern, Worcestershire, UK). Particle size and zeta potential were measured simultaneously, with each measurement repeated three times. Encapsulation efficacy of miRNA complexes were obtained by the RNA-quantification using Quant-it RiboGreen RNA Assay Kit following manufacturer’s instructions (Thermo Fisher Scientific, Waltham, MA, USA). The cut-off for RNA quantification was set at 200 pg.

### 4.3. Stability Studies of Lipoplexes

To evaluate the stability of lipoplexes under physiological conditions, lipoplexes at 10:1 and 16:1 ratios were incubated at 37 °C in PBS (pH 7.4) for 72 h. Particle size and zeta potential were measured at various time points (0, 4, 6, 8, 24, 28, 32, 48, 52, 56, and 72 h) using dynamic light scattering (DLS) and zeta potential analysis, respectively, to monitor changes in size and surface charge. Additionally, to simulate the effect of a biological environment, the lipoplexes were incubated with 10% FBS at 37 °C. The influence of serum proteins on lipoplex stability was assessed by measuring any changes in particle size or zeta potential, as serum proteins can form a protein corona that may alter lipoplex properties. These measurements provided insight into the lipoplex stability and their interactions with serum components.

### 4.4. Isolation of Peripheral Blood Mononuclear Cells (PBMCs) from CLE Patients

A total of five CLE patients were included in the study. Demographic characteristics are shown in Appendix A. Inclusion criteria for patients were: age ≥ 18 years, CLE Disease Area and Severity Index > 4, presence of a cutaneous lesion ≥ 3 cm, and no systemic or local therapy for at least 4 weeks before inclusion. Blood samples were collected using Vacutainer CPT tubes to isolate peripheral blood mononuclear cells (PBMCs) by density gradient centrifugation over Ficoll (BD Biosciences, Franklin Lakes, NJ, USA). Tubes were centrifuged at 3000 rpm for 30 min at room temperature (RT). The layer containing peripheral blood mononuclear cells (PBMCs) was clearly visible and was carefully collected using a pipette. Cells were washed twice with PBS and stored in liquid nitrogen until use. The study was approved by the Vall d’Hebrón Ethics Committee (PI21/01869, February 2022), and informed consent was obtained from all subjects prior to the study.

### 4.5. Cell Culture and In Vitro Experiments

Human epidermal adult keratinocytes (HEKas) were cultured in EpiLife serum-free media supplemented with Human Keratinocyte Growth Supplement (Life Technologies, Carlsbad, CA, USA). PBMCs isolated from CLE patients were cultured in complete RPMI media (RPMI with 10% FBS, 10% Pen/Strep, and 2 mM/L-glutamine; Gibco (Waltham, MA, USA), Life Technologies). Cells were seeded at a density of 1 × 10^5^ cells/well in a 24-well plate. At 70% confluence, HEKa cells in 1 mL of medium were stimulated with UVB radiation (25 mJ/cm^2^), and PBMCs were stimulated with PMA/ionomycin (10 ng/mL and 250 ng/mL, respectively) for 6 h. Subsequently, liposomes or lipoplexes were added at varying concentrations (5–40 μg/mL). After 24 h of incubation in minimal media, cells were analyzed using MTT assays, RT-qPCR, flow cytometry, or immunofluorescence.

### 4.6. Confocal Live-Cell Microscopy: Interaction Between Liposomes and HEKa or PBMCs

HEKa or PBMCs were seeded in a black 24-well plate (Ibidi, Gräfelfing, Germany) at a density of 1 × 10^5^ cells/mL and stained with the blue-fluorescent DNA stain Hoechst 33,342 (Thermo Fisher, Waltham, MA, USA) for live-cell analysis at 37 °C with 5% CO_2_. Liposomes or lipoplexes were labeled with the fluorescent dye DiO (Life Technologies, Carlsbad, CA, USA) during their preparation, following the manufacturer’s instructions. The DiO-labeled particles were added to cells at concentrations of 5 or 10 μg/mL. Images of selected sections (*n* = 5) were captured every 15 min over a 3-h period using a Zeiss LSM780 confocal microscope (Oberkochen, Germany). The Airyscan technique was applied to enhance signal quality and visualize cellular structures. Fluorescence images were overlaid with phase contrast (grey) images at 1024 × 1024 resolution, 16-bit depth, and 25× magnification. Image analysis was performed using ImageJ Fiji software (version 1.45, National Institutes of Health, Bethesda, MD, USA) [42] (details in Appendix A).

### 4.7. Immune 3D-Skin Organoid for Skin Penetration and Therapeutic Efficacy

T-Skin/Reconstructed Human Full Thickness Skin Models were purchased from EpiSkin (L’Oréal, Lyon, France) and seeded into inserts in six-well culture plates. Immune 3D-skin organoids were generated following the methodology described by Shin J.U. et al. [40].

CLE PBMCs (1 × 10^6^ cells) were seeded onto a collagen gel matrix and cultured overnight to allow cell adhesion and interaction with the extracellular matrix. The following day, the collagen gel with attached PBMCs was transferred into the T-skin organoid insert to establish a 3D immune-skin model that simulates the dermal environment. The immune 3D-skin model was cultured in a 1:1 mixture of complete RPMI media (Gibco, Life Technologies) and skin media (Episkin, Lyon, France) at 37 °C in a humidified incubator with 5% CO_2_ for 24 h. The organoid was subsequently exposed to UVB radiation (25 mJ/cm^2^). After 6 h, lipoplexes at a 10:1 ratio was added dropwise at a final concentration of 10 μg/mL.

For skin penetration studies, liposomes were pre-labeled with DiO and incubated for 4 h. For therapeutic efficacy assessments, liposomes or lipoplexes were incubated for 24 h, after which the organoids were either frozen in Optimal Cutting Temperature (OCT) compound or processed for RNA extraction. All experiments were performed three times.

### 4.8. MTT Assay

After cell stimulation and incubation with liposomes or lipoplexes, cell viability was assessed using the CyQUANT MTT Cell Viability Assay kit (Thermo Fisher Scientific, Waltham, MA, USA), according to the manufacturer’s instructions. Cells were seeded at a density of 1 × 10^4^ cells/well in a 96-well plate. The optical density (OD) was measured at 570 nm using a Varioskan Lux spectrophotometer (Thermo Fisher Scientific, Waltham, MA, USA).

### 4.9. miRNA and RNA Quantification by RT-qPCR

Total RNA was extracted from cultured cells and 3D-skin organoids using the RNeasy Mini Kit (Qiagen, Hilden, Germany). For miRNA and mRNA quantification, the MicroRNA Reverse Transcription Kit (Applied Biosystems, Foster City, CA, USA) and High-Capacity cDNA Reverse Transcription Kit (Applied Biosystems) were used, respectively. Gene expression was quantified using RT-qPCR with an ABI PRISM 7000 thermocycler (Applied Biosystems, Waltham, MA, USA) and TaqMan gene expression assays (FAM dye-labeled MGB probe, Applied Biosystems, Appendix A). More details are in the Appendix A.

### 4.10. Flow Cytometry

PBMCs were analyzed using a seven-color flow cytometer (LSRFortessa, BD Biosciences, Franklin Lakes, NJ, USA). For cell surface staining, conjugated monoclonal antibodies PE-CD3 and FITC-CD69 (BD Biosciences, Franklin Lakes, NJ, USA) were used. Data were analyzed with FlowJo software (version 10.10, FlowJo LLC, Ashland, OR, USA).

### 4.11. Immune 3D-Skin Organoid Immunofluorescence

Immunofluorescence was performed on 5-μm OCT-frozen skin sections. Primary and secondary antibodies used are listed in Appendix A. Staining was visualized using an Olympus IX71 (TH4-200) U-RFL-T microscope (Tokyo, Japan). Images were processed with ImageJ software (version 1.45, National Institutes of Health, Bethesda, MD, USA). Stained samples were evaluated semi-quantitatively by two blinded dermatopathologists (more details are in the Appendix A).

### 4.12. Immunofluorescence in Cultured Cells

After stimulation with UVB or IL-1α and incubation with the corresponding lipoplexes, cells were washed with PBS, fixed with 4% paraformaldehyde, and permeabilized with 0.1% Triton X-100. Primary antibodies (1:250 dilution) were incubated overnight, followed by incubation with secondary antibodies (1:1000 dilution) for 2 h at room temperature (Appendix A).

### 4.13. Western Blot Analysis

Western blot analysis was performed to assess NF-κB activation. Cells or skin organoids were lysed using RIPA buffer (Thermo Fisher Scientific, Waltham, MA, USA) supplemented with protease inhibitors (Roche, Basel, Switzerland). Protein concentrations were determined using the BCA assay (Thermo Fisher Scientific, Waltham, MA, USA). Equal amounts of protein (30 μg) were separated by SDS-PAGE and transferred to a nitrocellulose membrane. The membrane was blocked with 5% non-fat dry milk in TBST for 1 h at room temperature. Primary antibodies against NF-κB p65 (ab16502, Abcam, Cambridge, UK) and β-actin (mAbcam 8226, Abcam, Cambridge, UK) were incubated overnight at 4 °C at a 1:250 dilution. The membrane was then incubated with the secondary antibody, anti-human IgG HRP at 1:1000 (ab6858, Abcam, Cambridge, UK) for 2.5 h at room temperature. Signals were detected using SuperSignal West Pico Plus (Thermo Fisher Scientific, Waltham, MA, USA) and visualized using the Odyssey XF Imager System (LI-COR BioTech, Lincoln, NE, USA).

### 4.14. Statistical Analysis

Statistical analyses were performed using GraphPad Prism software (version 6.0.1). Results are presented as means ± SEM and were analyzed using Student’s *t*-tests, one-way or two-way ANOVA to identify differences between groups. A *p*-value < 0.05 was considered statistically significant. RT-qPCR data were analyzed using the 2^−ΔΔCt^ method to calculate fold changes.

## Figures and Tables

**Figure 1 ijms-26-02641-f001:**
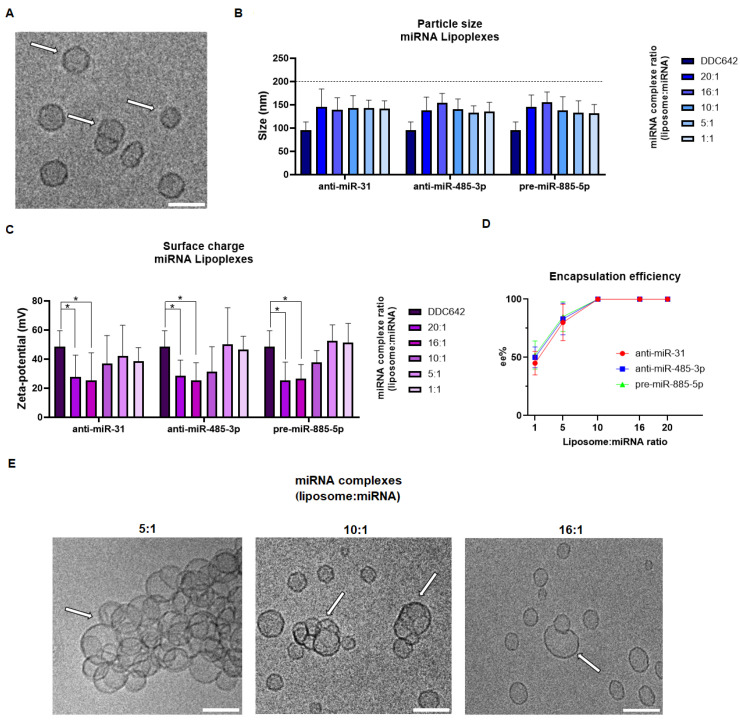
Characterization of DDC642 liposomes and their miRNA complexes. (**A**) Cryo-TEM image of DDC642 liposomes showing unilamellar vesicles with round shapes and elastic deformations (white arrows). Scale bar: 100 nm. (**B**) Particle size of miRNA lipoplexes (anti-miR-31, anti-miR-485-3p, and pre-miR-885-5p) at different liposome:miRNA ratios (20:1, 16:1, 10:1, 5:1, 1:1). A significant increase in size was observed upon complexation, with values remaining below 200 nm across all formulations. (**C**) Surface charge (zeta potential) of DDC642 liposomes and miRNA lipoplexes at various ratios. A significant reduction in zeta potential was observed for anti-miR-31, anti-miR-485-3p, and pre-miR-885-5p lipoplexes at 20:1 and 16:1 ratios. Statistical analysis was performed using two-way ANOVA, with comparisons made relative to DDC642 liposome. * *p* < 0.05. (**D**) Encapsulation efficiency (ee%) of miRNA in lipoplexes. High encapsulation efficiencies (~95%) were observed for all formulations except for the 1:1 ratio, which showed significantly lower efficiency (45–52%). (**E**) Cryo-TEM images of miRNA lipoplexes (anti-miR-31 shown as a representative example) at different liposome:miRNA ratios (5:1, 10:1, and 16:1). Aggregation and cluster-like structures were evident at the 5:1 ratio (white arrows), while more uniform mixed lipid bilayers with reduced aggregation were observed at 10:1 and 16:1 ratios (white arrows). Scale bar: 100 nm. All experiments were performed independently five times.

**Figure 2 ijms-26-02641-f002:**
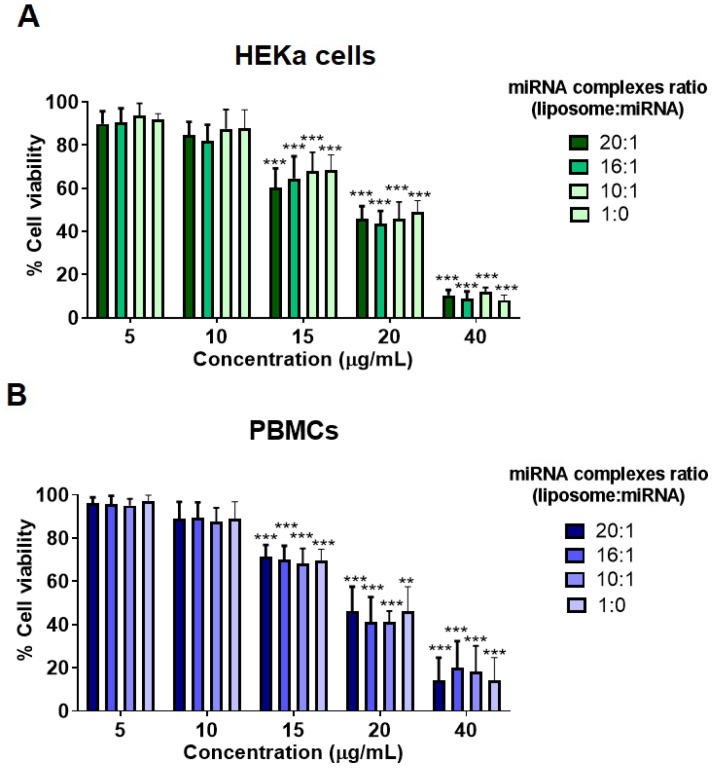
Cytotoxicity of miRNA liposome complexes in HEKa cells and PBMCs. Cytotoxic effects of miRNA liposome complexes at varying liposome-to-miRNA ratios (20:1, 16:1, 10:1, 1:0) in HEKa cells (**A**) and PBMCs (**B**). Cell viability was measured across a concentration range of 5–40 µg/mL. Statistical analysis was performed using two-way ANOVA, with comparisons made relative to the 5 µg/mL concentration. Experiments were performed in triplicate. ** *p* < 0.01, *** *p* < 0.001.

**Figure 3 ijms-26-02641-f003:**
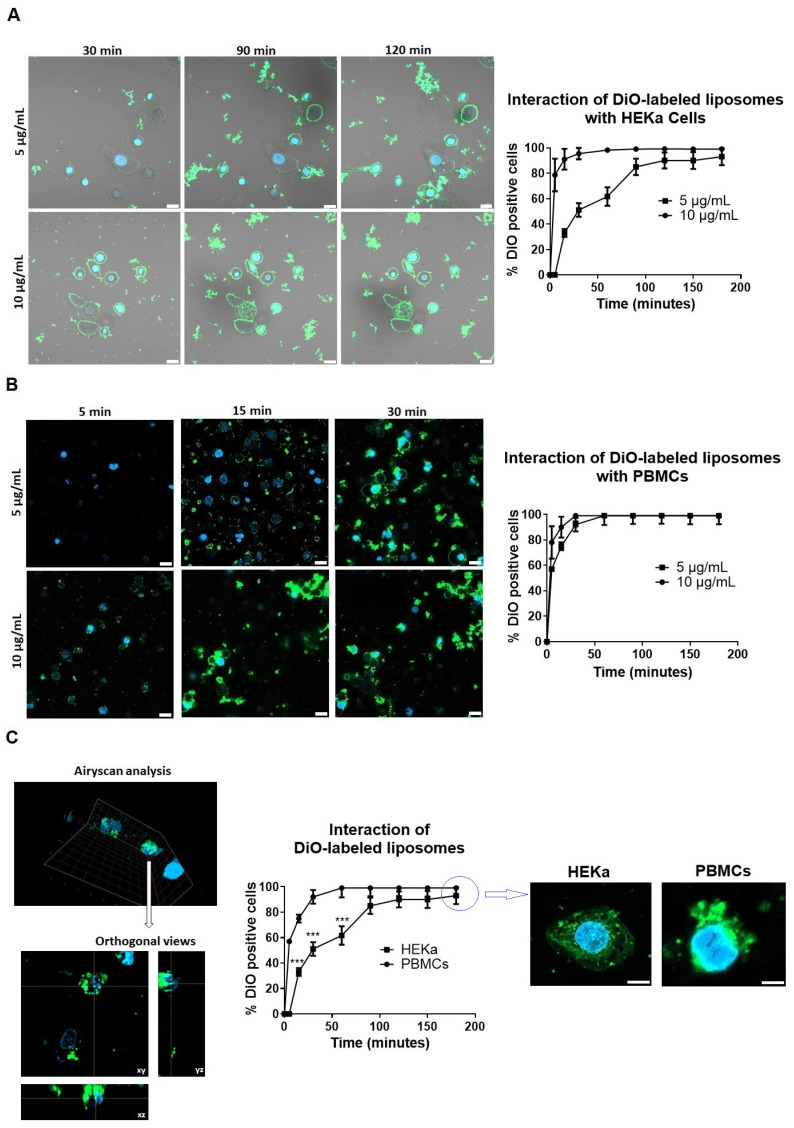
Time-dependent interaction of DiO-labeled liposomes with HEKa cells and PBMCs. (**A**,**B**) Confocal microscopy images illustrating the interaction of DiO-labeled liposomes with HEKa cells (**A**) and PBMCs (**B**) at concentrations of 5 µg/mL and 10 µg/mL. Liposomes were labeled with DiO (3,3′-dioctadecyloxacarbocyanine, perchlorate), emitting green fluorescence. The accompanying graph quantifies the percentage of DiO-positive cells over a 180-min period, based on the analysis of five distinct regions of interest, demonstrating a concentration-dependent interaction rate. Phase contrast images of HEKa cells were captured to observe cellular morphology. Scale bar = 20 µm. Images were obtained from five different regions per sample and experiments were performed in triplicate. (**C**) Comparison of the interaction between DiO-labeled liposomes and HEKa or PBMCs cells at concentration of 5 µg/mL. Confocal microscopy images were acquired using a Zeiss LSM980 microscope. Airyscan imaging was employed to enhance axial resolution. Orthogonal views were used to better highlight the interaction of liposomes with the cells. The graph compares the kinetics of the interaction, showing significantly higher interaction in PBMCs compared to HEKa cells until 90 minutes. Final images were recorded at 180 min, with scale bars of 10 µm for HEKa cells and 5 µm for PBMCs. Statistical analysis was performed using a two-way ANOVA test. *** *p* < 0.001.

**Figure 4 ijms-26-02641-f004:**
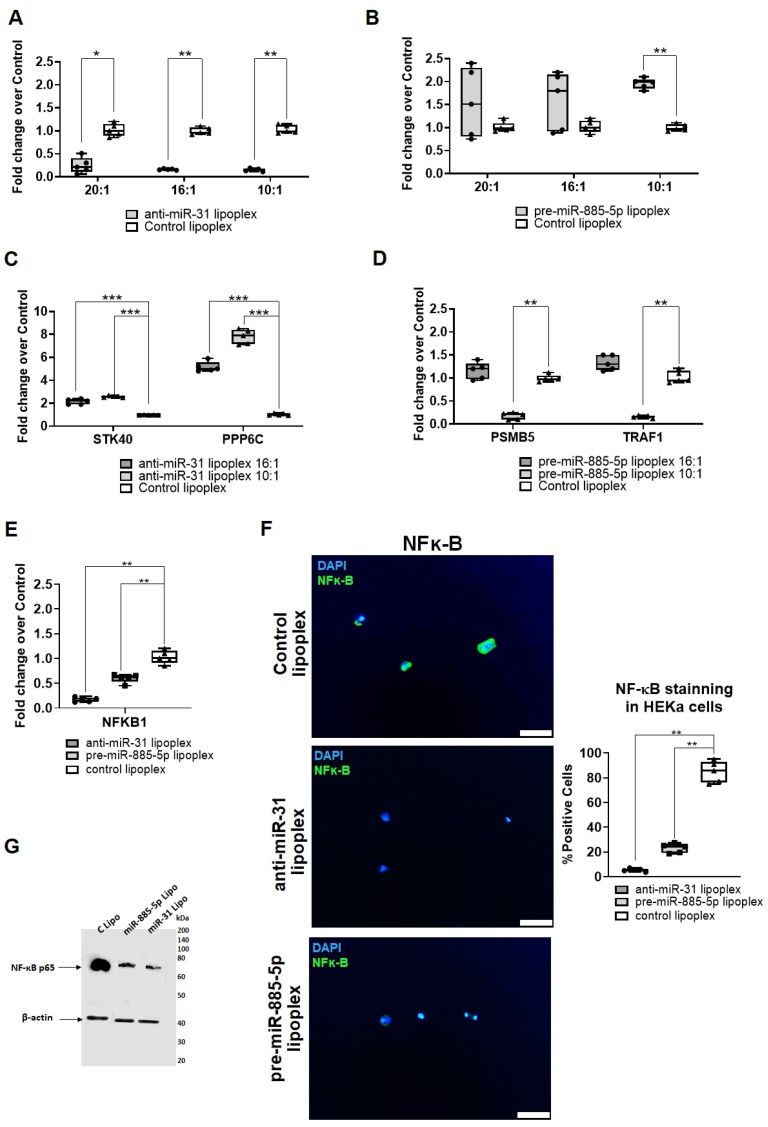
Effects of miR-31 and miR-885-5p modulation on NF-κB signaling and related gene expression in HEKa cells. (**A**,**B**) miRNA gene expression was quantified by RT-qPCR in UV-stimulated HEKa cells treated with anti-miR-31 (**A**) or pre-miR-885-5p (**B**) lipoplexes at varying ratios (20:1, 16:1, 10:1). Fold changes were calculated relative to control lipoplexes. Individual data points from five independent replicates are shown. Statistical analysis was performed using a one-way ANOVA test. * *p* < 0.05; ** *p* < 0.005. (**C**,**D**) Expression of miR-31 and miR-885-5p target genes after treatment with their respective lipoplexes at 10:1 or 16:1 ratios. Individual data points from five independent replicates are shown. Statistical significance was determined using a one-way ANOVA test compared to control lipoplexes.; ** *p* < 0.005; *** *p* < 0.001. (**E**) Expression of *NFKB1* in HEKa cells following treatment with lipoplexes at a 10:1 ratio. Data are presented as mean ± SD of three replicates. Statistical analysis was performed using a one-way ANOVA test compared to control lipoplexes. ** *p* < 0.005. (**F**) Immunofluorescence staining of NF-κB (green) and nuclei (DAPI, blue) in HEKa cells treated with control lipoplex, anti-miR-31 lipoplex, or pre-miR-885-5p lipoplex. Scale bar = 50 µm. Quantification of NF-κB-positive cells is shown. Statistical analysis was performed using a one-way ANOVA test compared to control lipoplexes. Individual data points from five independent replicates are shown. ** *p* < 0.005. (**G**) Western blot analysis of cell lysates detecting NF-κB p65 protein, with β-actin used as a loading control. Control lipoplexes containing scrambled miRNAs.

**Figure 5 ijms-26-02641-f005:**
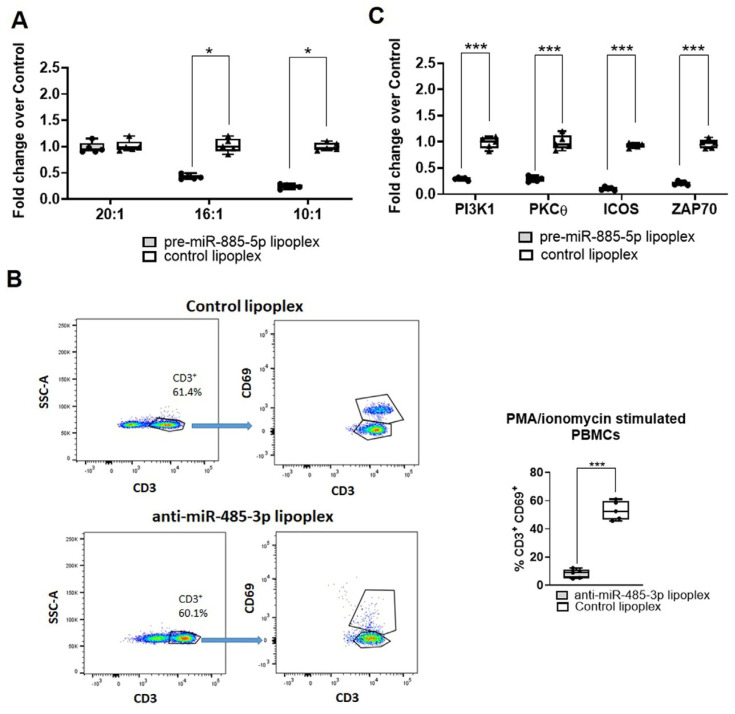
Impact of anti-miR-485-3p lipoplex treatment on PMA/ionomycin stimulated PBMCs. (**A**) Fold-change in miRNA gene expression (relative to control lipoplex) in PBMCs treated with anti-miR-485-3p lipoplex at varying ratios (20:1, 16:1, 10:1). miRNA expression was quantified by RT-qPCR. Results are presented as individual dots of five independent replicates. Statistical analysis was performed using a one-way ANOVA test. * *p* < 0.05. (**B**) Flow cytometric analysis of CD69 expression in CD3⁺ T cells after treatment with anti-miR-485-3p or control lipoplex. The gating strategy was performed by selecting CD3⁺ cells, followed by quantification of the percentage of CD69⁺CD3⁺ T cells. Representative flow cytometry plots (left) and quantification of the percentage of CD69⁺CD3⁺ T cells (right) are shown. Data are presented as mean ± SD of three replicates. Statistical analysis was performed using a Student’s *t*-test. *** *p* < 0.001. (**C**) Expression of *PI3K1*, *PKCθ*, *ICOS*, and *ZAP70*, key downstream regulators of T-cell activation, in PBMCs treated with anti-miR-485-3p lipoplex versus control lipoplex. Results are shown as fold-change relative to the control lipoplex, with data expressed as box plot with individual plots for five independent replicates. Statistical analysis was performed using a two-way ANOVA test. *** *p* < 0.001. Control lipoplexes containing scrambled miRNAs.

**Figure 6 ijms-26-02641-f006:**
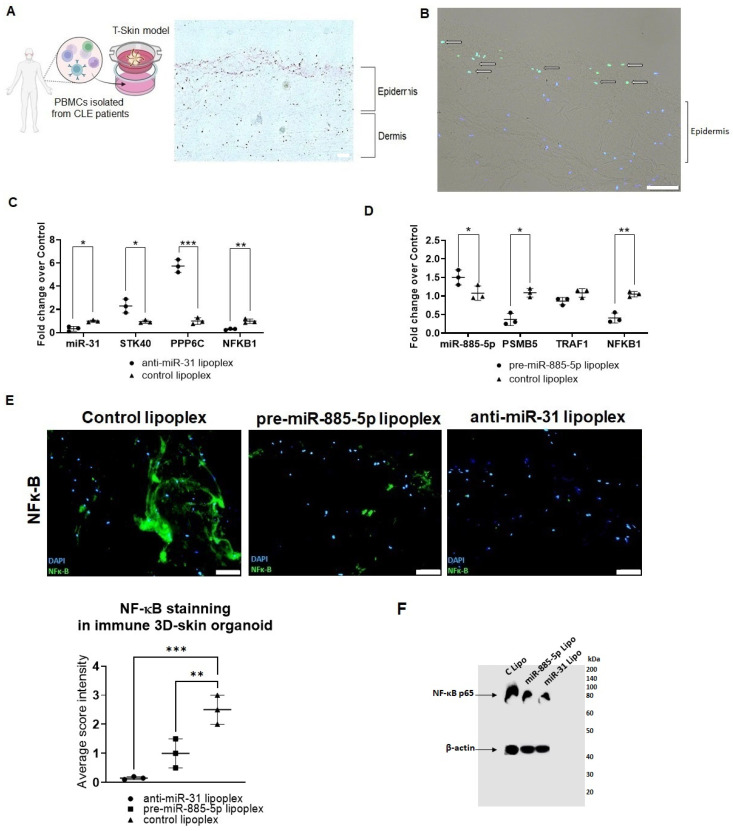
Evaluation of anti-miR-31 and pre-miR-885-5p lipoplexes in immune 3D-skin organoids using PBMCs from CLE patients. (**A**) Schematic representation of the experimental setup involving PBMCs from CLE patients cultured in a T-skin model, created using https://www.biorender.com/. A representative histological image shows the dermal-epidermal structure of the model. Scale bar: 100 µm. (**B**) Immunofluorescent staining of DiO-labeled lipoplexes incubated in CLE immune 3D-skin organoids. DAPI marks nuclei, and lipoplexes (green staining) are indicated by arrows. Scale bar: 100 µm. (**C**) Quantitative RT-PCR analysis of miR-31 and its downstream targets (*STK40*, *PPP6C*, and *NFKB1*) following anti-miR-31 lipoplex treatment in immune 3D-skin organoids. Fold changes were calculated relative to the control lipoplex treatment. Data represent triplicate experiments, with statistical analysis performed using a one-way ANOVA test. * *p* < 0.05, ** *p* < 0.005, *** *p* < 0.001. (**D**) Quantitative RT-PCR analysis of miR-885-5p and its downstream targets (*PSMB5*, *TRAF1*, and *NFKB1*) following pre-miR-885-5p lipoplex treatment in immune 3D-skin organoids. Fold changes were calculated relative to the control lipoplex treatment. Data represent triplicate experiments, with statistical analysis performed using a one-way ANOVA test. * *p* < 0.05, ** *p* < 0.005. (**E**) NF-κB immunofluorescent staining in 3D immune skin organoids treated with control lipoplex, anti-miR-31 lipoplex, or pre-miR-885-5p lipoplex. Green fluorescence represents NF-κB expression, and DAPI marks nuclei. Scale bar: 200 µm. The dot plot, representing three independent experiments, quantifies NF-κB intensity. Statistical analysis was performed using a one-way ANOVA test comparing each lipoplex to the control., ** *p* < 0.005, *** *p* < 0.001. (**F**) Western blot analysis to NF-κB p65 protein in 3D-skin models, with β-actin used as a loading control. Control lipoplexes containing scrambled miRNAs.

## Data Availability

Data is contained within the article and Appendix A.

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
