# Peer review of "Topical miRNA Delivery via Elastic Liposomal Formulation: A Promising Genetic Therapy for Cutaneous Lupus Erythematosus (CLE)"

_ijms, 2025, doi:10.3390/ijms26062641_

Round 1

Reviewer 1 Report

Comments and Suggestions for Authors

In this study, the authors encapsulated three microRNAs in liposomes and tested their efficacy in vitro and in an organoid skin model for lupus. The topic is timely and relevant. However, weaknesses include missing key controls, a lack of methodologic detail, use of a lipophilic dye to measure RNA entry, and unconvincing NF-KB data.

Major points

1. Fig 3, the DiO staining looks to be mostly on the surface or not associated with cells. Without a membrane marker, the authors cannot conclude there is any “liposome penetration”. Furthermore, the HeKa cells in the image look rounded and dead.

2. All of the microRNAs should be analyzed for the same targets. This will address the issue of potential off-targets. A scrambled miRNA would be a stronger control than empty liposomes.

3. Fig 3, unclear what timepoint Fig 3C is taken at. Furthermore, “nuclear” localization looks to be an artifact, because the staining pattern is not nuclear. More rigorous and careful quantitation is needed.

4. Fig 4, western blots to confirm NF-KB activation are needed. PCR is insufficient, and the images are unconvincing.

5. Fig 5, the % of viable CD3 cells for each group needs to be reported.

6. Fig 5, the flow cytometry needs additional rigor. The gating strategy needs to be included. For analysis of CD3 cells, the CD3 population needs to be gated on first, and percentages/MFI calculated from there. T cells further should be stimulated with a better T cell agonist than IL-1, such as anti-CD3/CD28, or PMA/ionomycin. IF these PBMCs are stimulated with IL-1, non-stimulated PBMCs must also be shown.

7. Fig 6, no keratinized layer is evident. It further does not appear the liposomes penetrated far or were taken up by cells.

8. Fig 6, non-UV controls are also needed. The UV phenotype appears weak. It is unclear there is any therapeutic affect in this model system.

9. Fig 6, western blotting for NF-KB is necessary.

10. The statistics need to be improved. Individual data points should be plotted instead of bar charts, the number of independent biological experiments need to be reported for all figure panels, and student’s t test should be avoided whenever there are multiple comparisons (eg Fig 4).

11. Methodologic details are sparse and insufficient to reproduce the study. For a short list of examples, line 409, the methods cut off, section 4.3 was ficoll used? Section 4.4 was UV exposure on cells in full medium, or after medium was aspirated? Section 4.5 no fixation is mentioned prior to DAPI staining (If there is no fixation, the authors are labeling dead cells, not the live ones). Temp for the later time series is not specified. Line 449, “T-cell-attached collagen” is unclear. Section 4.7 how many cells?

Minor points

1. What is the scale bar in Fig 1?

2. Figure 2 presentation would be improved by combining the data in panel A with panels B and C to make 2 bar charts. A “1:0” column can be added to each for control.

3. Fig 3C, “cytoplasmic” is mis-spelled. There are spelling errors in other figures, too.

4. Fig 3 is missing scale bars.

Author Response

In this study, the authors encapsulated three microRNAs in liposomes and tested their efficacy in vitro and in an organoid skin model for lupus. The topic is timely and relevant. However, weaknesses include missing key controls, a lack of methodologic detail, use of a lipophilic dye to measure RNA entry, and unconvincing NF-KB data.

Major points

  1. Fig 3, the DiO staining looks to be mostly on the surface or not associated with cells. Without a membrane marker, the authors cannot conclude there is any “liposome penetration”. Furthermore, the HeKa cells in the image look rounded and dead.

To better visualize cellular structures in keratinocytes, we repeated the in vivo analysis by overlaying fluorescence images with phase contrast images (Page 5, lines 165-166). The in vivo distribution of lipoplexes was monitored using Zeiss confocal microscopy, which generates 3D images of intracellular structures in living cells, allowing for the examination of dynamic cellular processes. Additionally, we employed the Airyscan technique to enhance signal detection by capturing light that would otherwise be rejected by the confocal pinhole. This approach improves the signal-to-noise ratio, providing high-resolution information. Orthogonal views were used to analyze images along all axes and confirm nuclear staining, as introduced in Figure 3C and discussed in both the manuscript (Page 15, lines 477-482) and supporting information. The methodology follows that described in the following reference:

  • Bucevičius, J., Kostiuk, G., Gerasimaitė, R., Gilat, T., & Lukinavičius, G. (2020). Enhancing the biocompatibility of rhodamine fluorescent probes by a neighbouring group effect. Chemical Science, 11(28), 7313-7323. https://doi.org/10.1039/d0sc02154g

Furthermore, Figure 3A has been improved by incorporating higher-quality images with phase contrast to better visualize cellular structures. This revision addresses concerns regarding the appearance of HEKa cells and the spatial association of DiO staining.

  1. All of the microRNAs should be analyzed for the same targets. This will address the issue of potential off-targets. A scrambled miRNA would be a stronger control than empty liposomes.

Following your suggestion, we analyzed the same targets for anti-miR-31 and pre-miR-885-5p, lipoplexes to address potential off-target effects. After conducting these analyses, we did not observe significant differences across the targets:

The corresponding graphs have been included in the supporting information (Figure S1) and are cited in the manuscript (Page 7, lines 198-200).

In the graphs, "control lipoplex" refers to liposomes containing scrambled miRNA: pre-miR miRNA negative control #2 (AM17111) and anti-miR miRNA inhibitor negative control #1 (AM17010). This information is provided in the methodology section (Page 13, lines 390-392). To further clarify, we have added a sentence in the results section (Page 7, lines 182-183). Additionally, we have ensured that this detail is specified in all figure legends.

  1. Fig 3, unclear what timepoint Fig 3C is taken at. Furthermore, “nuclear” localization looks to be an artifact, because the staining pattern is not nuclear. More rigorous and careful quantitation is needed.

The timepoint for Figure 3C is 180 minutes, the final timepoint of the analysis, and this information has been included in the legend and in the Figure 3. To enhance the visualization of nuclear localization, we used Aryscan images and orthogonal views:

 We have included this information in the manuscript (Page 15, lines 477-482) and in the supporting information.

Additionally, we have provided a detailed description of the methodology used to assess nuclear localization, including Airyscan image acquisition and quantification with ImageJ Fiji software, in the Methods section and supporting information.

  1. Fig 4, western blots to confirm NF-KB activation are needed. PCR is insufficient, and the images are unconvincing.

We agree that western blotting would provide more robust evidence of NF-κB activation. To strengthen our findings, we will include western blot analyses for phosphorylated p65, with β-actin as a loading control, alongside the RT-qPCR data:

  1. Fig 5, the % of viable CD3 cells for each group needs to be reported.

Following your suggestion, we have included the percentage of viable CD3 cells for each group in the Figure 5.

  1. Fig 5, the flow cytometry needs additional rigor. The gating strategy needs to be included. For analysis of CD3 cells, the CD3 population needs to be gated on first, and percentages/MFI calculated from there. T cells further should be stimulated with a better T cell agonist than IL-1, such as anti-CD3/CD28, or PMA/ionomycin. IF these PBMCs are stimulated with IL-1, non-stimulated PBMCs must also be shown.

We have included the gating strategy, where the CD3 population was gated first, and percentages were calculated accordingly. The experiment was repeated using PMA/ionomycin as a stimulation condition, and Figure 5 has been updated to reflect these results.

Additionally, we performed a control experiment with non-stimulated PBMCs, but no differences were observed between the control and anti-miR-485-3p lipoplex. These results have been included in the supporting information (Figure S3, Page 11 line 252).

  1. Fig 6, no keratinized layer is evident. It further does not appear the liposomes penetrated far or were taken up by cells.

We have updated Figure 6B by including both fluorescence and bright-field images to enhance visualization of liposome penetration. The new images clearly show that liposomes are primarily taken up by cells localized in the epidermis, addressing the concern regarding their limited penetration.

  1. Fig 6, non-UV controls are also needed. The UV phenotype appears weak. It is unclear there is any therapeutic effect in this model system.

We performed a non-UV control experiment, but no differences were observed between the lipoplexes when the model was not stimulated with UV. This information has been included in the manuscript (Page 10, lines 271-275).

  1. Fig 6, western blotting for NF-KB is necessary.

Following your suggestion, western blotting was performed and the results have been included in Figure 6F:

  1. The statistics need to be improved. Individual data points should be plotted instead of bar charts, the number of independent biological experiments need to be reported for all figure panels, and student’s t test should be avoided whenever there are multiple comparisons (eg Fig 4).

Following your suggestion, we have replaced bar charts with individual data points in Figures 4, 5, and 6 to improve data visualization. The number of independent biological experiments is now reported in the figure panels and detailed in the Methodology section. Additionally, we have reviewed all statistical analyses, applying one-way or two-way ANOVA tests for multiple comparisons, as indicated in the figure legends.

  1. Methodologic details are sparse and insufficient to reproduce the study. For a short list of examples, line 409, the methods cut off, section 4.3 was ficoll used? Section 4.4 was UV exposure on cells in full medium, or after medium was aspirated? Section 4.5 no fixation is mentioned prior to DAPI staining (If there is no fixation, the authors are labeling dead cells, not the live ones). Temp for the later time series is not specified. Line 449, “T-cell-attached collagen” is unclear. Section 4.7 how many cells?

We have enhanced the methodological details and included all the information you requested to improve the reproducibility of the study:

  • Line 409: The cut-off is set at 200 pg, as per the manufacturer’s instructions for the Quant-iT RiboGreen RNA Assay Kit.
  • Section 4.3: Blood samples were collected using Vacutainer CPT tubes to isolate peripheral blood mononuclear cells (PBMCs) by density gradient centrifugation over Ficoll.
  • Section 4.4: HEKa cells were stimulated with UVB radiation (25 mJ/cm²) in 1 mL of medium.
  • Section 4.5: Cells were stained with Hoechst 33342 for in vivo imaging analysis without fixation, and this has been clarified in the methods.
  • Temperature for later time series: The temperature has now been specified, and we have improved the description of the methodology.
  • Line 449: We revised the terminology and provided a clearer explanation regarding the T-cell attachment to collagen.
  • Section 4.7: The details of the number of cells used have been clarified.

We have carefully reviewed the entire methodology and added more detailed information in the supporting section to ensure the reproducibility of the experiments. Additionally, we have included more specific details regarding the Western blot procedures.

Minor points

  1. What is the scale bar in Fig 1?

The scale bar represents 100 µm, and we have included this information in the legend of Figure 1.

  1. Figure 2 presentation would be improved by combining the data in panel A with panels B and C to make 2 bar charts. A “1:0” column can be added to each for control.

We have combined the data from panels A, B, and C into two bar charts, including the "1:0" column as a control, following your suggestion. However, to clearly illustrate the effect of liposome concentration on % cell viability, we have retained Figure 2A in the supporting information (Figure S1).

  1. Fig 3C, “cytoplasmic” is mis-spelled. There are spelling errors in other figures, too.

We have corrected this spelling error and carefully reviewed all figures to ensure there are no additional typographical mistakes.

  1. Fig 3 is missing scale bars.

We have included the scale bars in Figure 3.

Reviewer 2 Report

Comments and Suggestions for Authors

The manuscript presents a well-structured and compelling study on the use of elastic liposomes as a novel genetic therapeutic strategy for cutaneous lupus erythematosus (CLE). The authors effectively describe the application of the DDC642 liposomal carrier in delivering miRNA mimics or inhibitors to target miR-31/miR-485-3p or upregulate miR-885-5p in cultured primary skin cells. The study is well-written, and the experimental design is logical and appropriate for the research question. However, a few minor revisions are recommended to enhance the clarity and scientific rigor of the manuscript.

Minor Revisions:

  1. Inclusion of Replication Numbers (n):

    • The authors should specify the number of replicates (n) for each experiment in the figure legends. This is crucial for ensuring statistical validity and reproducibility of the results.

  2. Dynamic Light Scattering (DLS) Analysis for Stability Assessment:

    • The authors have wisely used Dynamic Light Scattering (DLS) to evaluate the stability of the encapsulated liposomes over time in physiological conditions. It is suggested to include additional data showing the diameter change of the liposomes over time and, if possible, a corona formation study (Z potential in present of serum protein). This would provide further insight into the stability and integrity of the liposomal formulation.

Author Response

The manuscript presents a well-structured and compelling study on the use of elastic liposomes as a novel genetic therapeutic strategy for cutaneous lupus erythematosus (CLE). The authors effectively describe the application of the DDC642 liposomal carrier in delivering miRNA mimics or inhibitors to target miR-31/miR-485-3p or upregulate miR-885-5p in cultured primary skin cells. The study is well-written, and the experimental design is logical and appropriate for the research question. However, a few minor revisions are recommended to enhance the clarity and scientific rigor of the manuscript.

Minor Revisions:

  1. Inclusion of Replication Numbers (n):
    • The authors should specify the number of replicates (n) for each experiment in the figure legends. This is crucial for ensuring statistical validity and reproducibility of the results.

Following your suggestions, we have included all the number of replicates for each experiment in the figure legends.

  1. Dynamic Light Scattering (DLS) Analysis for Stability Assessment:
    • The authors have wisely used Dynamic Light Scattering (DLS) to evaluate the stability of the encapsulated liposomes over time in physiological conditions. It is suggested to include additional data showing the diameter change of the liposomes over time and, if possible, a corona formation study (Z potential in present of serum protein). This would provide further insight into the stability and integrity of the liposomal formulation.

We greatly appreciate the reviewer’s valuable suggestion regarding the stability assessment of the encapsulated liposomes. To address this point, we have included additional DLS data showing the change in the diameter of the liposomes over time under physiological conditions (37°C in PBS). The results demonstrate that the particle size remained relatively stable over an average period of 52 hours, suggesting moderate stability of the liposomal formulation.

Furthermore, to assess the formation of a protein corona, we have conducted Z-potential measurements in the presence of 10% FBS. The results show a non-significant shift in the Z-potential from +37.5 mV and +59.5 mV to +33.5 mV and +48.5 mV, respectively, suggesting limited adsorption of serum proteins onto the liposomal surface.

The new data have been incorporated into the Results section (Page 3, lines 109-113) and methodology section (Page 14, lines 432-441).

Round 2

Reviewer 1 Report

Comments and Suggestions for Authors

The authors addressed most of my concerns. The penetration of the liposomes still looks like the majority is on the surface. The staining in Fig 3 looks like the surface of the cells in the focal plane  above the cells visible by transmitted light. The bulk of the cell looks to be lying below the plane of focus. A surface marker is needed to conclude that the liposomes entered the cell. Or the text needs to be modified to match the data.

Author Response

We appreciate the reviewer’s thoughtful comment. We acknowledge that the current images may suggest that the liposomes are primarily localized at the cell surface. As we cannot conclusively determine the penetration of the liposomes without using a membrane surface marker, we have revised the manuscript accordingly. We have modified the text (Pages 5-6, Lines 155-176) to clarify that we observed liposome interaction, but not penetration. We also updated the figure legend to replace “penetration” with “interaction,” as the current data do not support the former conclusion. Additionally, we have removed the study of nuclear/cytoplasmic localization from the analysis. The discussion section has been updated to reflect these changes as well (Pages 15, Lines 372-376).
